# Equi-normalization of Neural Networks

**Pierre Stock[1,2], Benjamin Graham[1], Rémi Gribonval[2] and Hervé Jégou[1]**
[1]Facebook AI Research
[2]Univ Rennes, Inria, CNRS, IRISA
E-mail correspondance: `pstock@fb.com`

## Abstract

Modern neural networks are over-parametrized. In particular, each rectified linear hidden unit can be modified by a multiplicative factor by adjusting input and output weights, without changing the rest of the network. Inspired by the Sinkhorn-Knopp algorithm, we introduce a fast iterative method for minimizing the $\ell_2$ norm of the weights, equivalently the weight decay regularizer. It provably converges to a unique solution. Interleaving our algorithm with SGD during training improves the test accuracy. For small batches, our approach offers an alternative to batch- and group- normalization on CIFAR-10 and ImageNet with a ResNet-18.

## 1 Introduction

Deep Neural Networks (DNNs) have achieved outstanding performance across a wide range of empirical tasks such as image classification (Krizhevsky et al., 2012), image segmentation (He et al., 2017), speech recognition (Hinton et al., 2012a), natural language processing (Collobert et al., 2011) or playing the game of Go (Silver et al., 2017). These successes have been driven by the availability of large labeled datasets such as ImageNet (Russakovsky et al., 2015), increasing computational power and the use of deeper models (He et al., 2015b).

Although the expressivity of the function computed by a neural network grows exponentially with depth (Pascanu et al., 2013; Raghu et al., 2017; Telgarsky, 2016), in practice deep networks are vulnerable to both over- and underfitting (Glorot & Bengio, 2010; Krizhevsky et al., 2012; He et al., 2015b). Widely used techniques to prevent DNNs from overfitting include regularization methods such as weight decay (Krogh & Hertz, 1992), Dropout (Hinton et al., 2012b) and various data augmentation schemes (Krizhevsky et al., 2012; Simonyan & Zisserman, 2014; Szegedy et al., 2014; He et al., 2015b). Underfitting can occur if the network gets stuck in a local minima, which can be avoided by using stochastic gradient descent algorithms (Bottou, 2010; Duchi et al., 2011; Sutskever et al., 2013; Kingma & Ba, 2014), sometimes along with carefully tuned learning rate schedules (He et al., 2015b; Goyal et al., 2017).

Training deep networks is particularly challenging due to the vanishing/exploding gradient problem. It has been studied for Recurrent Neural networks (RNNs) (Hochreiter et al., 2001) as well as standard feedforward networks (He et al., 2015a; Mishkin & Matas, 2015). After a few iterations, the gradients computed during backpropagation become either too small or too large, preventing the optimization scheme from converging. This is alleviated by using non-saturating activation functions such as rectified linear units (ReLUs) (Krizhevsky et al., 2012) or better initialization schemes preserving the variance of the input across layers (Glorot & Bengio, 2010; Mishkin & Matas, 2015; He et al., 2015a). Failure modes that prevent the training from starting have been theoretically studied by Hanin & Rolnick (2018).

Two techniques in particular have allowed vision models to achieve "super-human" accuracy. Batch Normalization (BN) was developed to train Inception networks (Ioffe & Szegedy, 2015). It introduces intermediate layers that normalize the features by the mean and variance computed within the current batch. BN is effective in reducing training time, provides better generalization capabilities after training and diminishes the need for a careful initialization. Network architectures such as ResNet (He et al., 2015b) and DenseNet (Huang et al., 2016) use skip connections along with BN to improve the information flow during both the forward and backward passes.

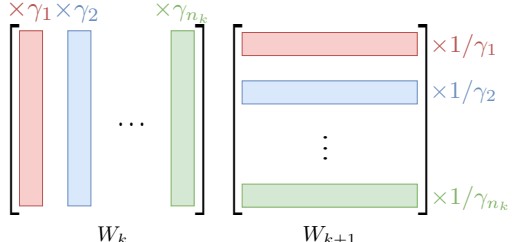

Figure 1: Matrices $W_k$ and $W_{k+1}$ are updated by multiplying the columns of the first matrix with rescaling coefficients. The rows of the second matrix are inversely rescaled to ensure that the product of the two matrices is unchanged. The rescaling coefficients are strictly positive to ensure functional equivalence when the matrices are interleaved with ReLUs. This rescaling is applied iteratively to each pair of adjacent matrices. In this paper, we address the more complex cases of biases, convolutions, max-pooling or skip-connections to be able to balance modern CNN architectures.

However, BN has some limitations. In particular, BN only works well with sufficiently large batch sizes (Ioffe & Szegedy, 2015; Wu & He, 2018). For sizes below 16 or 32, the batch statistics have a high variance and the test error increases significantly. This prevents the investigation of higher-capacity models because large, memory-consuming batches are needed in order for BN to work in its optimal range. In many use cases, including video recognition (Carreira & Zisserman, 2017) and image segmentation (He et al., 2017), the batch size restriction is even more challenging because the size of the models allows for only a few samples per batch. Another restriction of BN is that it is computationally intensive, typically consuming 20% to 30% of the training time. Variants such as Group Normalization (GN) (Wu & He, 2018) cover some of the failure modes of BN.

In this paper, we introduce a novel algorithm to improve both the training speed and generalization accuracy of networks by using their over-parameterization to regularize them. In particular, we focus on neural networks that are positive-rescaling equivalent (Neyshabur et al., 2015), *i.e.* whose weights are identical up to positive scalings and matching inverse scalings. The main principle of our method, referred to as *Equi-normalization* (ENorm), is illustrated in Figure 1 for the fully-connected case. We scale two consecutive matrices with rescaling coefficients that minimize the joint $\ell_p$ norm of those two matrices. This amounts to re-parameterizing the network under the constraint of implementing the same function. We conjecture that this particular choice of rescaling coefficients ensures a smooth propagation of the gradients during training.

A limitation is that our current proposal, in its current form, can only handle learned skip-connections like those proposed in type-C ResNet. For this reason, we focus on architectures, in particular ResNet18, for which the learning converges with learned skip-connection, as opposed to architectures like ResNet-50 for which identity skip-connections are required for convergence.

In summary,

- We introduce an iterative, batch-independent algorithm that re-parametrizes the network within the space of rescaling equivalent networks, thus preserving the function implemented by the network;

- We prove that the proposed Equi-normalization algorithm converges to a unique canonical parameterization of the network that minimizes the global $\ell_p$ norm of the weights, or equivalently, when $p = 2$, the weight decay regularizer;

- We extend ENorm to modern convolutional architectures, including the widely used ResNets, and show that the theoretical computational overhead is lower compared to BN ($\times 50$) and even compared to GN ($\times 3$);

- We show that applying one ENorm step after each SGD step outperforms both BN and GN on the CIFAR-10 (fully connected) and ImageNet (ResNet-18) datasets.

- Our code is available at `https://github.com/facebookresearch/enorm`.

The paper is organized as follows. Section 2 reviews related work. Section 3 defines our Equi-normalization algorithm for fully-connected networks and proves the convergence. Section 4 shows how to adapt ENorm to convolutional neural networks (CNNs). Section 5 details how to employ ENorm for training neural networks and Section 6 presents our experimental results.

## 2   RELATED WORK

This section reviews methods improving neural network training and compares them with ENorm. Since there is a large body of literature in this research area, we focus on the works closest to the proposed approach. From early works, researchers have noticed the importance of normalizing the input of a learning system, and by extension the input of any layer in a DNN (LeCun et al., 1998). Such normalization is applied either to the weights or to the activations. On the other hand, several strategies aim at better controlling the geometry of the weight space with respect to the loss function. Note that these research directions are not orthogonal. For example, explicitly normalizing the activations using BN has smoothing effects on the optimization landscape (Santurkar et al., 2018).

**Normalizing activations.** Batch Normalization (Ioffe & Szegedy, 2015) normalizes the activations by using statistics computed along the batch dimension. As stated in the introduction, the dependency on the batch size leads BN to underperform when small batches are used. Batch Renormalization (BR) (Ioffe, 2017) is a follow-up that reduces the sensitivity to the batch size, yet does not completely alleviate the negative effect of small batches. Several batch-independent methods operate on other dimensions, such as Layer Normalization (channel dimension) (Ba et al., 2016) and Instance-Normalization (sample dimension) (Ulyanov et al., 2016). Parametric data-independent estimation of the mean and variance in every layer is investigated by Arpit et al. (2016). However, these methods are inferior to BN in standard classification tasks. More recently, Group Normalization (GN) (Wu & He, 2018), which divides the channels into groups and normalizes independently each group, was shown to effectively replace BN for small batch sizes on computer vision tasks.

**Normalizing weights.** Early weight normalization techniques only served to initialize the weights before training (Glorot & Bengio, 2010; He et al., 2015a). These methods aim at keeping the variance of the output activations close to one along the whole network, but the assumptions made to derive these initialization schemes may not hold as training evolves. More recently, Salimans & Kingma (2016) propose a polar-like re-parametrization of the weights to disentangle the direction from the norm of the weight vectors. Note that Weight Norm (WN) does require mean-only BN to get competitive results, as well as a greedy layer-wise initialization as mentioned in the paper.

**Optimization landscape.** Generally, in the parameter space, the loss function moves quickly along some directions and slowly along others. To account for this anisotropic relation between the parameters of the model and the loss function, *natural gradient* methods have been introduced (Amari, 1998). They require storing and inverting the $N \times N$ *curvature matrix*, where $N$ is the number of network parameters. Several works approximate the inverse of the curvature matrix to circumvent this problem (Pascanu & Bengio, 2013; Marceau-Caron & Ollivier, 2016; Martens & Grosse, 2015). Another method called Diagonal Rescaling (Lafond et al., 2017) proposes to tune a particular re-parametrization of the weights with a block-diagonal approximation of the inverse curvature matrix. Finally, Neyshabur et al. (2015) propose a rescaling invariant path-wise regularizer and use it to derive Path-SGD, an approximate steepest descent with respect to the path-wise regularizer.

**Positioning.** Unlike BN, Equi-normalization focuses on the weights and is independent of the concept of batch. Like Path-SGD, our goal is to obtain a balanced network ensuring a good back-propagation of the gradients, but our method explicitly re-balances the network using an iterative algorithm instead of using an implicit regularizer. Moreover, ENorm can be readily adapted to the convolutional case whereas Neyshabur et al. (2015) restrict themselves to the fully-connected case. In addition, the theoretical computational complexity of our method is much lower than the overhead introduced by BN or GN (see Section 5). Besides, WN parametrizes the weights in a polar-like manner, $w = g \times v/|v|$, where $g$ is a scalar and $v$ are the weights, thus it does not balance the network but individually scales each layer. Finally, Sinkhorn's algorithm aims at making a single matrix doubly stochastic, while we balance a product of matrices to minimize their global norm.

## 3   EQUI-NORMALIZATION

We first define Equi-normalization in the context of simple feed forward networks that consist of two operators: linear layers and ReLUs. The algorithm is inspired by Sinkhorn-Knopp and is designed to balance the energy of a network, *i.e.*, the $\ell_p$-norm of its weights, while preserving its function. When not ambiguous, we may denote by *network* a weight parametrization of a given network architecture.

### 3.1 NOTATION AND DEFINITIONS

We consider a network with $q$ linear layers, whose input is a row vector $x \in \mathbf{R}^{n_0}$. We denote by $\sigma$ the point-wise ReLU activation. For the sake of exposition, we omit a bias term at this stage. We recursively define a simple fully connected feedforward neural network with $L$ layers by $y_0 = x$,

$$y_k = \sigma\left(y_{k-1}W_k\right), \qquad k \in [\![1, q-1]\!], \tag{1}$$

and $y_q = y_{q-1}W_q$. Each linear layer $k$ is parametrized by a matrix $W_k \in \mathbf{R}^{n_{k-1} \times n_k}$. We denote by $f_\theta(x) = y_q$ the function implemented by the network, where $\theta$ is the concatenation of all the network parameters. We denote by $\mathcal{D}(n)$ the set of diagonal matrices of size $n \times n$ for which all diagonal elements are strictly positive and by $I_n$ the identity matrix of size $n \times n$.

**Definition 1.** *$\theta$ and $\tilde{\theta}$ are functionally equivalent if, for all $x \in \mathbf{R}^{n_0}$, $f_\theta(x) = f_{\tilde{\theta}}(x)$.*

**Definition 2.** *$\theta$ and $\tilde{\theta}$ are rescaling equivalent if, for all $k \in [\![1, q-1]\!]$, there exists a rescaling matrix $D_k \in \mathcal{D}(n_k)$ such that, for all $k \in [\![1, q-1]\!]$,*

$$\widetilde{W_k} = D_{k-1}^{-1}W_k D_k \tag{2}$$

*with the conventions that $D_0 = I_{n_0}$ and $D_q = I_{n_q}$. This amounts to positively scaling all the incoming weights and inversely scaling all the outgoing weights for every hidden neuron.*

Two weights vectors $\theta$ and $\tilde{\theta}$ that are rescaling equivalent are also functionally equivalent (see Section 3.5 for a detailed derivation). Note that a functional equivalence class is not entirely described by rescaling operations. For example, permutations of neurons inside a layer also preserve functional equivalence, but do not change the gradient. In what follows our objective is to seek a canonical parameter vector that is rescaling equivalent to a given parameter vector. The same objective under a *functional equivalence* constraint is beyond the scope of this paper, as there exist degenerate cases where functional equivalence does not imply rescaling equivalence, even up to permutations.

### 3.2 OBJECTIVE FUNCTION: CANONICAL REPRESENTATION

Given a network $f_\theta$ and $p > 0$, we define the $\ell_p$ norm of its weights as $\ell_p(\theta) = \sum_{k=1}^{q}\|W_k\|_p^p$. We are interested in minimizing $\ell_p$ inside an equivalence class of neural networks in order to exhibit a unique canonical element per equivalence class. We denote the *rescaling coefficients* within the network as $d_k \in (0, +\infty)^{n_k}$ for $k \in [\![1, q-1]\!]$ or as diagonal matrices $D_k = \mathrm{Diag}(d_k) \in D(n_k)$. We denote $\delta = (d_1, \ldots, d_{q-1}) \in \mathbf{R}^n$, where $n$ is the number of hidden neurons. Fixing the weights $\{W_k\}$, we refer to $\{D_{k-1}^{-1}W_k D_k\}$ as the *rescaled weights*, and seek to minimize their $\ell_p$ norm as a function of the rescaling coefficients:

$$\varphi(\delta) = \sum_{k=1}^{q} \left\| D_{k-1}^{-1}W_k D_k \right\|_p^p. \tag{3}$$

### 3.3 COORDINATE DESCENT: ENORM ALGORITHM

We formalize the ENorm algorithm using the framework of block coordinate descent. We denote by $W[:, j]$ (resp. $W_k[i, :]$) the $j^{\text{th}}$ column (resp. $i^{\text{th}}$ row) of a matrix $W_k$. In what follows we assume that each hidden neuron is connected to at least one input *and* one output neuron. ENorm generates a sequence of rescaling coefficients $\delta^{(r)}$ obtained by the following steps.

(1) **Initialization.** Define $\delta^{(0)} = (1, \ldots, 1)$.

(2) **Iteration.** At iteration $r$, consider layer $\ell \in [\![1, q-1]\!]$ such that $\ell - 1 \equiv r \bmod q - 1$ and define

$$\begin{cases} d_k^{(r+1)} = d_\ell^{(r)} \text{ if } k \neq \ell \\ d_\ell^{(r+1)} = \underset{t \in (0, +\infty)^{n_\ell}}{\mathrm{argmin}} \; \varphi\left(d_1^{(r)}, \ldots, d_{\ell-1}^{(r)}, t, d_{\ell+1}^{(r)}, \ldots, d_{q-1}^{(r)}\right). \end{cases}$$

Denoting $uv$ the coordinate-wise product of two vectors and $u/v$ for the division, we have

$$d_\ell^{(r+1)}[i] = \sqrt{\frac{\left\| W_{\ell+1}[i, :]d_{\ell+1}^{(r)} \right\|_p}{\left\| W_\ell[:, i]/d_{\ell-1}^{(r)} \right\|_p}}. \tag{4}$$

---

**Algorithm 1: Pseudo-code of Equi-normalization**

---

**Input:** Current layer weights $W_1, \ldots, W_q$, number of cycles $C$, choice of norm $p$
**Output:** Balanced layer weights
```
// Perform T ENorm cycles
```
**for** $t = 1 \ldots T$ **do**
    ```// Iterate through the layers```
    **for** $k = 2 \ldots q$ **do**
        $L[j] \longleftarrow \|W_{k-1}[:, j]\|_p$ for all $j \in \mathbf{R}^{n_k}$
        $R[i] \longleftarrow \|W_k[i, :]\|_p$   for all $i \in \mathbf{R}^{n_k}$
        $D_{k-1} \longleftarrow \text{Diag} \sqrt{R/L}$
        $W_{k-1} \longleftarrow W_{k-1} D_{k-1}$
        $W_k \longleftarrow (D_{k-1})^{-1} W_k$

---

**Algorithm and pseudo-code.** Algorithm 1 gives the pseudo-code of ENorm. By convention, one ENorm cycle balances the entire network once from $\ell = 1$ to $\ell = q - 1$. See Appendix A for illustrations showing the effect of ENorm on network weights.

### 3.4 CONVERGENCE

We now state our main convergence result for Equi-normalization. The proof relies on a coordinate descent Theorem by Tseng (2001) and can be found in Appendix B.1. The main difficulty is to prove the uniqueness of the minimum of $\varphi$.

**Theorem 1.** *Let $p > 0$ and $(\delta^{(r)})_{r \in \mathbf{N}}$ be the sequence of rescaling coefficients generated by ENorm from the starting point $\delta^{(0)}$ as described in Section 3.3. We assume that each hidden neuron is connected to at least one input* and *one output neuron. Then,*

*(1) **Convergence**. The sequence of rescaling coefficients $\delta^{(r)}$ converges to $\delta^*$ as $r \to +\infty$. As a consequence, the sequence of rescaled weights also converges;*

*(2) **Minimum global $\ell_p$ norm**. The rescaled weights after convergence minimize the global $\ell_p$ norm among all rescaling equivalent weights;*

*(3) **Uniqueness**. The minimum is unique, i.e. $\delta^*$ does not depend on the starting point $\delta^{(0)}$.*

### 3.5 HANDLING BIASES – FUNCTIONAL EQUIVALENCE

In the presence of biases, the network is defined as $y_k = \sigma(y_{k-1} W_k + b_k)$ and $y_q = y_{q-1} W_q + b_k$ where $b_k \in \mathbf{R}^{n_k}$. For rescaling-equivalent weights satisfying (2), in order to preserve the input-output function, we define matched rescaling equivalent biases $\widetilde{b}_k = b_k D_k$. In Appendix B.2, we show by recurrence that for every layer $k$,

$$\widetilde{y}_k = y_k D_k, \tag{5}$$

where $\widetilde{y}_k$ (resp. $y_k$) is the intermediary network function associated with the weights $\widetilde{W}$ (resp. $W$). In particular, $\widetilde{y}_q = y_q$, *i.e.* rescaling equivalent networks are functionally equivalent. We also compute the effect of applying ENorm on the gradients in the same Appendix.

### 3.6 ASYMMETRIC SCALING

Equi-normalization is easily adapted to introduce a depth-wise penalty on each layer. We consider the weighted loss $\ell_{p,(c_1,\ldots,c_q)}(\theta) = \sum_{k=1}^q c_k \|W_k\|^p$. This amounts to modifiying the rescaling coefficients as

$$\tilde{d}_\ell^{(r+1)}[i] = d_\ell^{(r+1)}[i] \left(c_{k+1}/c_k\right)^{1/2p}. \tag{6}$$

In Section 6, we explore two natural ways of defining $c_k$: $c_k = c^{p(q-k)}$ (**uniform**) and $c_k = 1/(n_{k-1} n_k)$ (**adaptive**). In the uniform setup, we penalize layers exponentially according to their depth: for instance, values of $c$ larger than 1 increase the magnitude of the weights at the end of the network. In the adaptive setup, the loss is weighted by the size of the matrices.

## 4 EXTENSION TO CNNS

We now extend ENorm to CNNs, by focusing on the typical ResNet architecture. We briefly detail how we adapt ENorm to convolutional or max-pooling layers, and then how to update an elementary block with a skip-connection. We refer the reader to Appendix C for a more extensive discussion. Sanity checks of our implementation are provided in Appendix E.1.

### 4.1 CONVOLUTIONAL LAYERS

Figure 2 explains how to rescale two consecutive convolutional layers. As detailed in Appendix C, this is done by first properly reshaping the filters to 2D matrices, then performing the previously described rescaling on those matrices, and then reshaping the matrices back to convolutional filters. This matched rescaling does preserve the function implemented by the composition of the two layers, whether they are interleaved with a ReLU or not. It can be applied to any two consecutive convolutional layers with various stride and padding parameters. Note that when the kernel size is 1 in both layers, we recover the fully-connected case of Figure 1.

### 4.2 MAX-POOLING

The MaxPool layer operates per channel by computing the maximum within a fixed-size kernel. We adapt Equation (5) to the convolutional case where the rescaling matrix $D_k$ is applied to the channel dimension of the activations $y_k$. Then,

$$\max\left(\widetilde{y}_k\right) = \max\left(y_k D_k\right) = \max\left(y_k\right) D_k. \tag{7}$$

Thus, the activations before and after the MaxPool layer have the same scaling and the functional equivalence is preserved when interleaving convolutional layers with MaxPool layers.

### 4.3 SKIP-CONNECTION

We now consider an elementary block of a ResNet-18 architecture as depicted in Figure 3. In order to maintain functional equivalence, we only consider ResNet architectures of type C as defined in (He et al., 2015b), where all shortcuts are learned $1 \times 1$ convolutions. As detailed in Appendix C, rescaling two consecutive blocks requires (a) to define the structure of the rescaling process, *i.e.* where to insert the rescaling coefficients and (b) a formula for computing those rescaling coefficients.

## 5 TRAINING PROCEDURE: EQUI-NORMALIZATION & SGD

**ENorm & SGD.** As detailed in Algorithm 2, we balance the network periodically after updating the gradients. By design, this does not change the function implemented by the network but will yield different gradients in the next SGD iteration. Because this re-parameterization performs a jump in the parameter space, we update the momentum using Equation (17) and the same matrices $D_k$ as those used for the weights. The number of ENorm cycles after each SGD step is an hyperparameter and by default we perform one ENorm cycle after each SGD step. In Appendix D, we also explore a method to jointly learn the rescaling coefficients and the weights with SGD, and report corresponding results.

**Computational advantage over BN and GN.** Table 1 provides the number of elements (weights or activations) accessed when normalizing using various techniques. Assuming that the complexity (number of operations) of normalizing is proportional to the number of elements and assuming all techniques are equally parallelizable, we deduce that our method (ENorm) is theoretically 50 times faster than BN and 3 times faster than GN for a ResNet-18. In terms of memory, ENorm requires no extra-learnt parameters, but the number of parameters learnt by BN and GN is negligible (4800 for a ResNet-18 and 26,650 for a ResNet-50). We implemented ENorm using a tensor library; to take full advantage of the theoretical reduction in compute would require an optimized CUDA kernel.

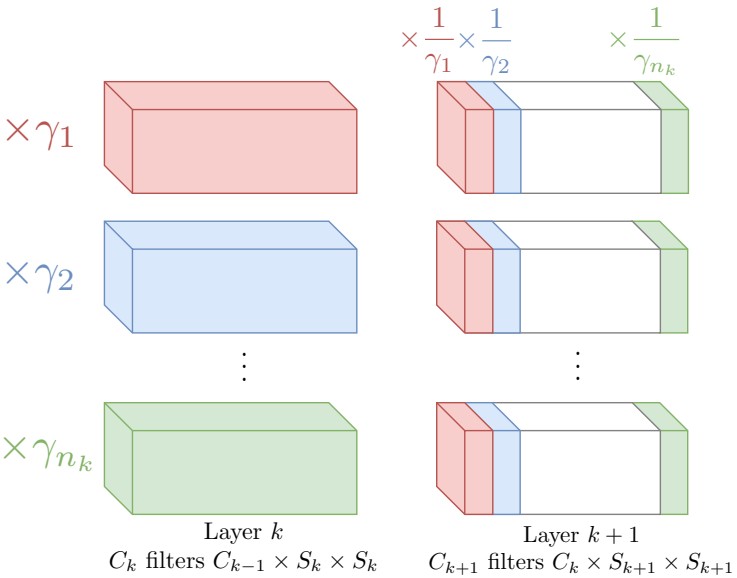

Figure 2: Rescaling the weights of two consecutive convolutional layers that preserves the function implemented by the CNN. Layer $k$ scales channel number $i$ of the input activations by $\gamma_i$ and layer $k+1$ cancels this scaling with the inverse scalar so that the activations after layer $k+1$ are unchanged.

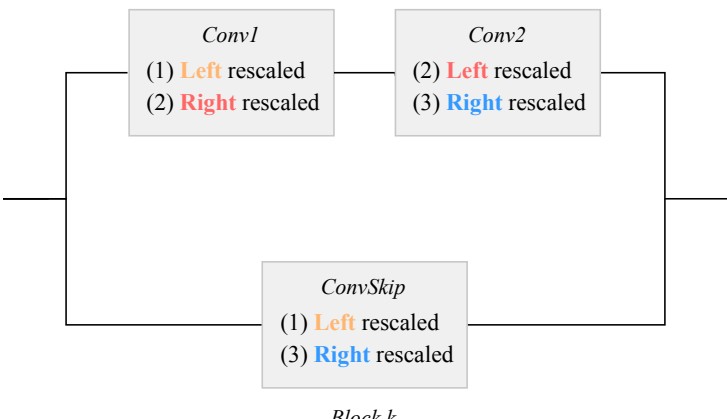

Figure 3: Rescaling an elementary block within a ResNet-18 consists of 3 steps. **(1)** Conv1 and ConvSkip are left-rescaled using the rescaling coefficients between blocks $k-1$ and $k$; **(2)** Conv1 and Conv2 are rescaled as two usual convolutional layers; **(3)** Conv2 and ConvSkip are right-rescaled using the rescaling coefficients between blocks $k$ and $k+1$. Identical colors denote the same rescaling coefficients $D$. Coefficients between blocks are rescaled as detailed in Section C.2.

## 6 EXPERIMENTS

We analyze our approach by carrying out experiments on the MNIST and CIFAR-10 datasets and on the more challenging ImageNet dataset. ENorm will refer to Equi-normalization with $p = 2$.

### 6.1 MNIST AUTO-ENCODER

**Training.** We follow the setup of Desjardins et al. (2015). Input data is normalized by subtracting the mean and dividing by standard deviation. The encoder has the structure FC(784, 1000)-ReLU-FC(1000, 500)-ReLU-FC(500, 250)-ReLU-FC(250, 30) and the decoder has the symmetric structure. We use He's initialization for the weights. We select the learning rate in $\{0.001, 0.01, 0.1\}$

**Algorithm 2:** Training with Equi-normalization

**Input:** Initialized network
**Output:** Trained network
**for** iteration $= 1 \ldots N$ **do**
  Update learning rate $\eta$
  Compute forward pass
  Compute backward pass
  Perform SGD step and update weights
  Perform one ENorm cycle using matrices $D_k$
  Update momentum buffers with the same $D_k$

| Model | ENorm | BN ($B$=256) | GN ($B$=16) |
|---|---|---|---|
| ResNet-18 | 12 | 636 | 40 |
| ResNet-50 | 30 | 2,845 | 178 |

Table 1: Number of elements that are accessed during normalization (in million of activations/parameters, rounded to the closest million). For BN and GN, we choose the typical batch size B used for training.

and decay it linearly to zero. We use a batch size of 256 and SGD with no momentum and a weight decay of 0.001. For path-SGD, our implementation closely follows the original paper (Neyshabur et al., 2015) and we set the weight decay to zero. For GN, we cross-validate the number of groups among $\{5, 10, 20, 50\}$. For WN, we use BN as well as a greedy layer-wise initialization as described in the original paper.

**Results.** While ENorm alone obtains competitive results compared to BN and GN, ENorm + BN outperforms all other methods, including WN + BN. Note that here ENorm refers to Enorm using the adaptive $c$ parameter as described in Subsection 3.6, whereas for ENorm + BN we use the uniform setup with $c = 1$. We perform a parameter study for different values and setups of the asymmetric scaling (uniform and adaptive) in Appendix E.2. Without BN, the adaptive setup outperforms all other setups, which may be due to the strong bottleneck structure of the network. With BN, the dynamic is different and the results are much less sensitive to the values of $c$. Results without any normalization and with Path-SGD are not displayed because of their poor performance.

## 6.2 CIFAR-10 FULLY CONNECTED

**Training.** We first experiment with a basic fully-connected architecture that takes as input the flattened image of size 3072. Input data is normalized by subtracting mean and dividing by standard deviation independently for each channel. The first linear layer is of size $3072 \times 500$. We then consider $p$ layers $500 \times 500$, $p$ being an architecture parameter for the sake of the analysis. The last classification is of size $500 \times 10$. The weights are initialized with He's scheme. We train for 60 epochs using SGD with no momentum, a batch size of 256 and weight decay of $10^{-3}$. Cross validation is used to pick an initial learning rate in $\{0.0005, 0.001, 0.005, 0.01, 0.05, 0.1\}$. Path-SGD, GN and WN are learned as detailed in Section 6.1. All results are the average test accuracies over 5 training runs.

**Results.** ENorm alone outperforms both BN and GN for any depth of the network. ENorm + BN outperforms all other methods, including WN + BN, by a good margin for more than $p = 11$ intermediate layers. Note that here ENorm as well as ENorm + BN refers to ENorm in the uniform setup with $c = 1.2$. The results of the parameter study for different values and setups of the asymmetric scaling are similar to those of the MNIST setup, see Appendix E.2.

## 6.3 CIFAR-10 FULLY CONVOLUTIONAL

**Training.** We use the CIFAR-NV architecture as described by Gitman & Ginsburg (2017). Images are normalized by subtracting mean and dividing by standard deviation independently for each channel. During training, we use $28 \times 28$ random crops and randomly flip the image horizontally. At test time, we use $28 \times 28$ center crops. We split the train set into one training set (40,000 images) and one validation set (10,000 images). We train for 128 epochs using SGD and an initial learning rate cross-validated on a held-out set among $\{0.01, 0.05, 0.1\}$, along with a weight decay of 0.001. The learning rate is then decayed quadratically to $10^{-4}$. We compare various batch sizes together with the use of momentum (0.9) or not. The weights are initialized with He's scheme. In order to stabilize the training, we employ a BatchNorm layer at the end of the network after the FC layer for the Baseline and ENorm cases. For GN we cross-validate the number of groups among $\{4, 8, 16, 32, 64\}$. All results are the average test accuracies over 5 training runs.

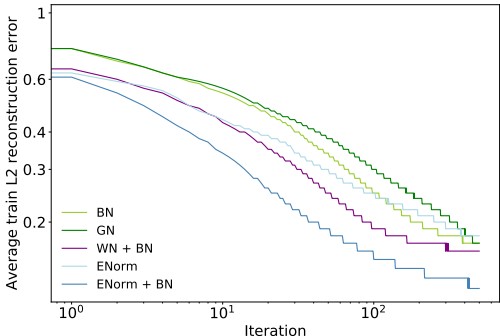

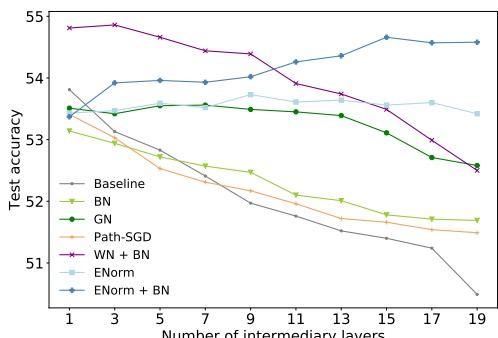

Figure 4: MNIST auto-encoder results (lower is better).

Figure 5: CIFAR-10 fully-connected results (higher is better).

| Method | Average train $L_2$ error |
|---|---|
| Baseline | 0.542 |
| BN | 0.171 |
| GN | 0.171 |
| WN + BN | 0.162 |
| ENorm | 0.179 |
| ENorm + BN | **0.102** |

| Method | Test top 1 accuracy |
|---|---|
| Baseline | 88.94 |
| BN | 90.32 |
| GN | 90.36 |
| WN + BN | 90.50 |
| ENorm | 89.31 |
| ENorm + BN | **91.35** |

Table 2: MNIST auto-encoder results (lower is better).

Table 3: CIFAR-10 fully convolutional results (higher is better).

**Results.** See Table 3. ENorm + BN outperforms all other methods, including WN + BN, by a good margin. Note that here ENorm refers to ENorm in the uniform setup with the parameter $c = 1.2$ whereas ENorm + BN refers to the uniform setup with $c = 1$. A parameter study for different values and setups of the asymmetric scaling can be found in Appendix E.2.

## 6.4 IMAGENET

This dataset contains 1.3M training images and 50,000 validation images split into 1000 classes. We use the ResNet-18 model with type-C learnt skip connections as described in Section 4.

**Training.** Our experimental setup closely follows that of GN (Wu & He, 2018). We train on 8 GPUs and compute the batch mean and standard deviation per GPU when evaluating BN. We use the Kaiming initialization for the weights (He et al., 2015a) and the standard data augmentation scheme of Szegedy et al. (2014). We train our models for 90 epochs using SGD with a momentum of 0.9. We adopt the linear scaling rule for the learning rate (Goyal et al., 2017) and set the initial learning rate to $0.1B/256$ where the batch size $B$ is set to $32, 64, 128$, or $256$. As smaller batches lead to more iterations per epoch, we adopt a similar rule and adopt a weight decay of $w = 10^{-4}$ for $B = 128$ and 256, $w = 10^{-4.5}$ for $B = 64$ and $w = 10^{-5}$ for $B = 32$. We decay the learning rate quadratically (Gitman & Ginsburg, 2017) to $10^{-5}$ and report the median error rate on the final 5 epochs. When using GN, we set the number of groups $G$ to 32 and did not cross-validate this value as prior work (Wu & He, 2018) reports little impact when varying $G$ from 2 to 64. In order for the training to be stable and faster, we added a BatchNorm at the end of the network after the FC layer for the Baseline and ENorm cases. The batch mean and variance for this additional BN are shared across GPUs. Note that the activation size at this stage of the network is $B \times 1000$, which is a negligible overhead (see Table 1).

**Results.** We compare the Top 1 accuracy on a ResNet-18 when using no normalization scheme, (Baseline), when using BN, GN and ENorm (our method). In both the Baseline and ENorm settings, we add a BN at the end of the network as described in 6.3. The results are reported in Table 4. The

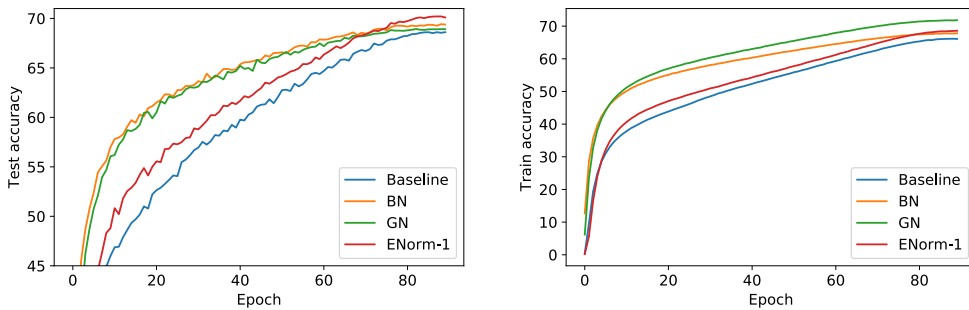

Figure 6: ResNet-18 results on the ImageNet dataset, batch size 64.

performance of BN decreases with small batches, which concurs with prior observations (Wu & He, 2018). Our method outperforms GN and BN for batch sizes ranging from 32 to 128. GN presents stable results across batch sizes. Note that values of $c$ different from 1 did not yield better results. The training curves for a batch size of 64 are presented in Figure 6. While BN and GN are faster to converge than ENorm, our method achieves better results after convergence in this case. Note also that ENorm overfits the training set less than BN and GN, but more than the Baseline case.

| Batch size | 32 | 64 | 128 | 256 |
|---|---|---|---|---|
| Baseline | 66.20 | 68.60 | 69.20 | 69.58 |
| BN | 68.01 | 69.38 | 70.83 | **71.37** |
| GN | 68.94 | 68.90 | 70.69 | 70.64 |
| ENorm-1 (ours) | **69.70** | **70.10** | **71.03** | 71.14 |

Table 4: ResNet-18 results on the ImageNet dataset (test accuracy).

## 6.5 LIMITATIONS

We applied ENorm to a deeper (ResNet-50), but obtained unsatisfactory results. We observed that learnt skip-connections, even initialized to identity, make it harder to train without BN, even with careful layer-wise initialization or learning rate warmup. This would require further investigation.

## 7 CONCLUDING REMARKS

We presented Equi-normalization, an iterative method that balances the energy of the weights of a network while preserving the function it implements. ENorm provably converges towards a unique equivalent network that minimizes the $\ell_p$ norm of its weights and it can be applied to modern CNN architectures. Using ENorm during training adds a much smaller computational overhead than BN or GN and leads to competitive performances in the FC case as well as in the convolutional case.

**Discussion.** While optimizing an unbalanced network is hard (Neyshabur et al., 2015), the criterion we optimize to derive ENorm is likely not optimal regarding convergence or training properties. These limitations suggest that further research is required in this direction.

## ACKNOWLEDGMENTS

We thank the anonymous reviewers for their detailed feedback, which helped us to significantly improve the paper's clarity and the experimental validation. We also thank Timothée Lacroix, Yann Ollivier and Léon Bottou for useful feedback on various aspects of this paper.

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

## APPENDIX A    ILLUSTRATION OF THE EFFECT OF EQUI-NORMALIZATION

We first apply ENorm to one randomly initialized fully connected network comprising 20 intermediary layers. All the layers have a size $500 \times 500$ and are initialized following the Xavier scheme. The network has been artificially unbalanced as follows: all the weights of layer 6 are multiplied by a factor 1.2 and all the weights of layer 12 are multiplied by 0.8, see Figure 7. We then iterate our ENorm algorithm on the network, without training it, to see that it naturally re-balances the network, see Figure 8.

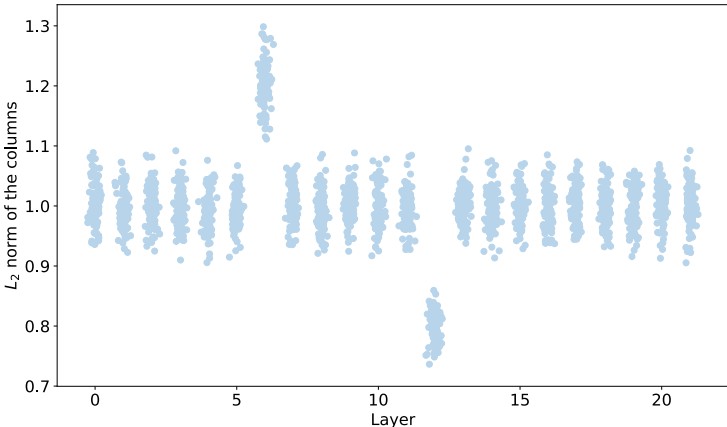

Figure 7: Energy of the network ($\ell_2$-norm of the weights), before ENorm. Each dot represents the norm of one column in the layer's weight matrix.

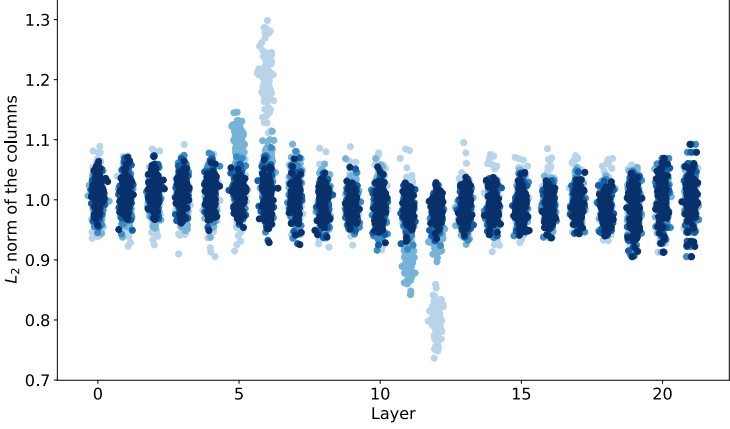

Figure 8: Energy of the network through successive ENorm iterations (without training). One color denotes one iteration. The darker the color, the higher the iteration number.

## APPENDIX B    PROOFS

### B.1    CONVERGENCE OF EQUI-NORMALIZATION

We now prove Theorem 1. We use the framework of block coordinate descent and we first state a consequence of a theorem of Tseng (2001) [Theorem 4.1][1].

---

[1]Note that what Tseng denotes as *stationary point* in his paper is actually defined as a point where the directional derivative is positive along every possible direction, *i.e.* a local minimum.

**Theorem 2.** *Let $D \subset \mathbf{R}^n$ be an open set and $f : D \to \mathbf{R}$ a real function of $B$ block variables $x_b \in \mathbf{R}^{n_b}$ with $\sum_{b=1}^{B} n_b = n$. Let $x^{(0)}$ be the starting point of the coordinate descent algorithm and $X$ the level set $X = \left\{ x \mid f(x) \leq f\left(x^{(0)}\right) \right\}$. We make the following assumptions:*

*(1) $f$ is differentiable on $D$ ;*

*(2) $X$ is compact ;*

*(3) for each $x \in D$, each block coordinate function $f_\ell : t \to f(x_1, \ldots, x_{\ell-1}, t, x_{\ell+1}, \ldots, x_B)$, where $2 \leq \ell \leq B - 1$, has at most one minimum.*

*Under these assumptions, the sequence $(x^{(r)})_{r \in \mathbf{N}}$ generated by the coordinate descent algorithm is defined and bounded. Moreover, every cluster point of $(x^{(r)})_{r \in \mathbf{N}}$ is a local minimizer of $f$.*

STEP 1. We apply Theorem 2 to the function $\varphi$. This is possible because all the assumptions are verified as shown below. Recall that

$$\varphi(\delta) = \sum_{k=1}^{q} \left\| D_{k-1}^{-1} W_k D_k \right\|_p^p = \sum_{k=1}^{q} \sum_{i,j} \left| \frac{d_k[j]}{d_{k-1}[i]} W_k[i,j] \right|^p . \tag{8}$$

**Assumption (1).** $\varphi$ is differentiable on the open set $D = (0, +\infty)^n$.

**Assumption (2).** $\varphi \to +\infty$ when $\|\delta\| \to +\infty$. Let $\delta$ such that $\varphi(\delta) < M^p$, $M > 1$. Let us show by induction that for all $k \in [\![1, q-1]\!]$, $\|d_k\|_\infty < (CM)^k$, where $C = \max(C_0, 1)$ and

$$C_0 = \max_{W_k[i,j] \neq 0} \left( \frac{1}{|W_k[i,j]|} \right) \tag{9}$$

- For the first hidden layer, index $k = 1$. By assumption, every hidden neuron is connected at least to one neuron in the input layer. Thus, for every $j$, there exists $i$ such that $W_1[i,j] \neq 0$. Because $\varphi(\delta) < M^p$ and $d_0[i] = 1$ for all $i$,

$$(d_1[j])^p \, |W_1[i,j]|^p = \left( \frac{d_1[j]}{d_0[i]} \right)^p |W_1[i,j]|^p < M^p \tag{10}$$

  Thus $\|d_1\|_\infty < CM$.

- For some hidden layer, index $k$. By assumption, every hidden neuron is connected at least to one neuron in the previous layer. Thus, for every $j$, there exists $i$ such that $W_k[i,j] \neq 0$. Because $\varphi(\delta) < M$,

$$\left( \frac{d_k[j]}{d_{k-1}[i]} \right)^p |W_k[i,j]|^p < M^p \tag{11}$$

  Using the induction hypothesis, we get $\|d_k\|_\infty < (CM)^k$.

Thus, $\|\delta\|_\infty < (MC)^q$ because $MC > 1$. By contraposition, $\varphi \to +\infty$.

Thus, there exists a ball $B$ such that $\delta \notin B$ implies $\varphi(\delta) > \varphi(\delta^0)$. Thus, $\delta \in X$ implies that $x \in B$ and $X \subset B$ is bounded. Moreover, $X$ is closed because $\varphi$ is continuous thus $X$ is compact and Assumption (2) is satisfied.

**Assumption (3).** We next note that

$$\varphi_l(t) = \varphi \left( d_1^{(r)}, \ldots, d_{\ell-1}^{(r)}, t, d_{\ell+1}^{(r)}, \ldots, d_{q-1}^{(r)} \right) \tag{12}$$

has a unique minimum as shown in Section 3.3, see Equation (4). The existence and uniqueness of the minimum comes from the fact that each hidden neuron is connected to at least one input *and* one output neuron, thus all the row and column norms of the hidden weight matrices $W_k$ are non-zero, as well as the column (resp. row) norms or $W_1$ (resp. $W_q$).

STEP 2. We prove that $\varphi$ has at most one stationary point on $D$ under the assumption that each hidden neuron is connected *either* to an input neuron *or* to an output neuron, which is weaker than the general assumption of Theorem 1.

We first introduce some definitions. We denote the set of all neurons in the network by $V$. Each neuron $\nu \in V$ belongs to a layer $k \in [\![0, q]\!]$ and has an index $i \in [\![1, n_k]\!]$ in this layer. Any edge $e$ connects some neuron $i$ at layer $k - 1$ to some neuron $j$ at layer $k$, $e = (k, i, j)$. We further denote by $H$ the set of hidden neurons $\nu$ belonging to layers $q \in [\![1, q-1]\!]$. We define E as the set of edges whose weights are non-zero, *i.e.*

$$E = \{(k, i, j) \mid W_{i,j}^{(k)} \neq 0\}. \tag{13}$$

For each neuron $\nu$, we define $\operatorname{prev}(\nu)$ as the neurons connected to $\nu$ that belong to the previous layer.

We now show that $\varphi$ has at most one stationary point on $D$. Directly computing the gradient of $\varphi$ and solving for zeros happens to be painful or even intractable. Thus, we define a change of variables as follows. We define $h$ as

$$h: (0, +\infty)^H \to \mathbf{R}^H$$
$$\delta \mapsto \log(\delta)$$

We next define the shift operator $S: \mathbf{R}^V \to \mathbf{R}^E$ such that, for every $x \in \mathbf{R}^V$,

$$S(x) = (\nu - \nu')_{\nu, \nu' \in V \text{ s.t. } \nu' \in \operatorname{prev}(\nu)}$$

and the padding operator $P$ as

$$P: \mathbf{R}^H \to \mathbf{R}^V$$

$$x \mapsto y \text{ where } \begin{cases} y_\nu = 0 & \text{if } \nu \in V \backslash H; \\ y_\nu = x_\nu & \text{otherwise.} \end{cases}$$

We define the extended shift operator $S_H = S \circ P$. We are now ready to define our change of variables. We define $\chi = \psi \circ S_H$ where

$$\psi: \mathbf{R}^E \to \mathbf{R}$$

$$x \mapsto \sum_{e \in E} \exp(px_e) |w_e|^p$$

and observe that

$$\varphi = \chi \circ h \tag{14}$$

so that its differential satisfies

$$[D\varphi](\delta) = [D\chi](h(\delta))[Dh](\delta). \tag{15}$$

Since $h$ is a $\mathcal{C}^\infty$ diffeomorphism, its differential $[Dh](\delta)$ is invertible for any $\delta$. It follows that $[D\varphi](\delta) = 0$ if, and only if, $[D\chi](h(\delta)) = 0$. As $\chi$ is the composition of a strictly convex function, $\psi$, and a linear injective function, $S_H$ (proof after Step 3), it is strictly convex. Thus $\chi$ has at most one stationary point, which concludes this step.

STEP 3. We prove that the sequence $\delta^{(r)}$ converges. Step 1 implies that the sequence $\delta^{(r)}$ is bounded and has at least one cluster point, as $f$ is continuous on the compact $X$. Step 2 implies that the sequence $\delta^{(r)}$ has at most one cluster point. We then use the fact that any bounded sequence with exactly one cluster point converges to conclude the proof.

$S$ IS INJECTIVE. Let $x \in \ker S_H$. Let us show by induction on the hidden layer index $k$ that for every neuron $\nu$ at layer $k$, $x_\nu = 0$.

- $k = 1$. Let $\nu$ be a neuron at layer 1. Then, there exists a path coming from an input neuron to $\nu_0$ through edge $e_1$. By definition, $P(x)_{\nu_0} = 0$ and $P(x)_\nu = x_\nu$, hence $S_H(x)_{e_1} = x_\nu - 0$. Since $S_H(x) = 0$ it follows that $x_\nu = 0$.
- $k \to k + 1$. Same reasoning using the fact that $x_{\nu_k} = 0$ by the induction hypothesis.

The case where the path goes from neuron $\nu$ to some output neuron is similar.

### B.2 Functional Equivalence

We show (5) by induction that for every layer $k$, i.e.,

$$\widetilde{y}_k = y_k D_k,$$

where $\widetilde{y}_k$ (resp. $y_k$) is the intermediary network function associated with weights $\widetilde{W}$ (resp. $W$). This holds for $k = 0$ since $D_0 = I_{n_0}$ by convention. If the property holds for some $k < q - 1$, then by (2) we have $\widetilde{y}_k \widetilde{W}_{k+1} = y_k D_k \widetilde{W}_{k+1} = y_k W_{k+1} D_{k+1}$ hence, since $\widetilde{b}_{k+1} = b_{k+1} D_{k+1}$,

$$\widetilde{y}_{k+1} = \sigma\left(\widetilde{y}_k \widetilde{W}_{k+1} + \widetilde{b}_{k+1}\right) = \sigma\left(y_k W_{k+1} D_{k+1} + b_{k+1} D_{k+1}\right)$$

$$= \sigma\left(y_k W_{k+1} + b_{k+1}\right) D_{k+1} = y_{k+1} D_{k+1}.$$

The same equations hold for $k = q - 1$ without the non-linearity $\sigma$.

Using the chain rule and denoting by $\ell$ the loss of the network, for every layer $k$, using (5), we have

$$\frac{\partial \ell}{\partial \widetilde{y}_k} = \frac{\partial \ell}{\partial y_k} (D_k)^{-1}. \tag{16}$$

Similarly, we obtain

$$\frac{\partial \ell}{\partial \widetilde{W}_k} = D_{k-1} \frac{\partial \ell}{\partial W_k} (D_k)^{-1} \qquad \text{and} \qquad \frac{\partial \ell}{\partial \widetilde{b}_k} = \frac{\partial \ell}{\partial b_k} (D_k)^{-1}. \tag{17}$$

Equation (17) will be used to update the momentum (see Section 5) and Equation (2) for the weights.

## Appendix C   Extension of ENorm to CNNs

### C.1 Convolutional layers

Let us consider two consecutive convolutional layers $k$ and $k + 1$, without bias. Layer $k$ has $C_k$ filters of size $C_{k-1} \times S_k \times S_k$, where $C_{k-1}$ is the number of input features and $S_k$ is the kernel size. This results in a weight tensor $T_k$ of size $C_k \times C_{k-1} \times S_k \times S_k$. Similarly, layer $k + 1$ has a weight matrix $T_{k+1}$ of size $C_{k+1} \times C_k \times S_{k+1} \times S_{k+1}$. We then perform axis-permutation and reshaping operations to obtain the following 2D matrices:

$$M_k \quad \text{of size } (C_{k-1} \times S_k \times S_k) \times C_k; \tag{18}$$

$$M_{k+1} \text{ of size } C_k \times (C_{k+1} \times S_{k+1} \times S_{k+1}). \tag{19}$$

For example, we first reshape $T_k$ as a 2D matrix by collapsing its last 3 dimensions, then transpose it to obtain $M_k$. We then jointly rescale those 2D matrices using rescaling matrices $D_k \in \mathcal{D}(k)$ as detailed in Section 3 and perform the inverse axis permutation and reshaping operations to obtain a *right-rescaled* weight tensor $\widetilde{T}_k$ and a *left-rescaled* weight tensor $\widetilde{T}_{k+1}$. See Figure 2 for an illustration of the procedure. This matched rescaling does preserve the function implemented by the composition of the two layers, whether they are interleaved with a ReLU or not. It can be applied to any two consecutive convolutional layers with various stride and padding parameters. Note that when the kernel size is 1 in both layers, we recover the fully-connected case of Figure 1.

### C.2 Skip-connection

We now consider an elementary block of a ResNet-18 architecture as depicted in Figure 3. In order to maintain functional equivalence, we only consider ResNet architectures of type C as defined in (He et al., 2015b), where all shortcuts are learned $1 \times 1$ convolutions.

**Structure of the rescaling process.** Let us consider a ResNet block $k$. We first left-rescale the *Conv1* and *ConvSkip* weights using the rescaling coefficients calculated between blocks $k - 1$ and $k$. We then rescale the two consecutive layers *Conv1* and *Conv2* with their own specific rescaling coefficients, and finally right-rescale the *Conv2* and *ConvSkip* weights using the rescaling coefficients calculated between blocks $k$ and $k + 1$.

**Computation of the rescaling coefficients.** Two types of rescaling coefficients are involved, namely those between two convolution layers inside the same block and those between two blocks. The rescaling coefficients between the *Conv1* and *Conv2* layers are calculated as explained in Section 4.1. Then, in order to calculate the rescaling coefficients between two blocks, we compute *equivalent block weights* to deduce rescaling coefficients.

We empirically explored some methods to compute the equivalent weight of a block using electrical network analogies. The most accurate method we found is to compute the equivalent weight of the *Conv1* and *Conv2* layers, *i.e.*, to express the succession of two convolution layers as only one convolution layer denoted as *ConvEquiv* (series equivalent weight), and in turn to express the two remaining parallel layers *ConvEquiv* and *ConvSkip* again as a single convolution layer (parallel equivalent weight). It is not possible to obtain series of equivalent weights, in particular when the convolution layers are interleaved with ReLUs. Therefore, we approximate the equivalent weight as the parallel equivalent weight of the *Conv1* and *ConvSkip* layers.

## Appendix D    Implicit Equi-normalization

In Section 3, we defined an iterative algorithm that minimizes the global $\ell_p$ norm of the network

$$\ell_2(\theta, \delta) = \sum_{k=1}^{q} \left\| D_{k-1}^{-1} W_k D_k \right\|_p^p. \tag{20}$$

As detailed in Algorithm 2, we perform alternative SGD and ENorm steps during training. We now derive an implicit formulation of this algorithm that we call *Implicit Equi-normalization*. Let us fix $p = 2$. We denote by $\mathcal{C}(f_\theta(x), y)$ the cross-entropy loss for the training sample $(x, y)$ and by $\ell_2(\theta, \delta)$ the weight decay regularizer (20). The loss function of the network writes

$$L(\theta, \delta) = \mathcal{C}(f_\theta(x), y) + \lambda \ell_2(\theta, \delta) \tag{21}$$

where $\lambda$ is a regularization parameter. We now consider both the weights *and* the rescaling coefficients as learnable parameters and we rely on automatic differentiation packages to compute the derivatives of $L$ with respect to the weights and to the rescaling coefficients. We then simply train the network by performing iterative SGD steps and updating all the learnt parameters. Note that by design, the derivative of $\mathcal{C}$ with respect to any rescaling coefficient is zero. Although the additional overhead of implicit ENorm is theoretically negligible, we observed an increase of the training time of a ResNet-18 by roughly 30% using PyTorch 4.0 (Paszke et al., 2017). We refer to Implicit Equi-normalization as ENorm-Impl and to Explicit Equi-normalization as ENorm.

We performed early experiments for the CIFAR10 fully-connected case. ENorm-Impl performs generally better than the baseline but does not outperform explicit ENorm, in particular when the network is deep. We follow the same experimental setup than previously, except that we additionally cross-validated $\lambda$. We also initialize all the rescaling coefficients to one.. Recall that ENorm or ENorm denotes explicit Equi-normalization while ENorm-Impl denotes Implicit Equi-normalization. We did not investigate learning the weights and the rescaling coefficients at different speeds (*e.g.* with different learning rates or momentum). This may explain in part why ENorm-Impl generally underperforms ENorm in those early experiments.

## Appendix E    Experiments

We perform sanity checks to verify our implementation and give additional results.

### E.1    Sanity checks

We apply our Equi-normalization algorithm to a ResNet architecture by integrating all the methods exposed in Section 4. We perform three sanity checks before proceeding to experiments. First, we randomly initialize a ResNet-18 and verify that it outputs the same probabilities before and after balancing. Second, we randomly initialize a ResNet-18 and perform successive ENorm cycles (without any training) and observe that the $\ell_2$ norm of the weights in the network is decreasing and then converging, as theoretically proven in Section 3, see Figure 9.

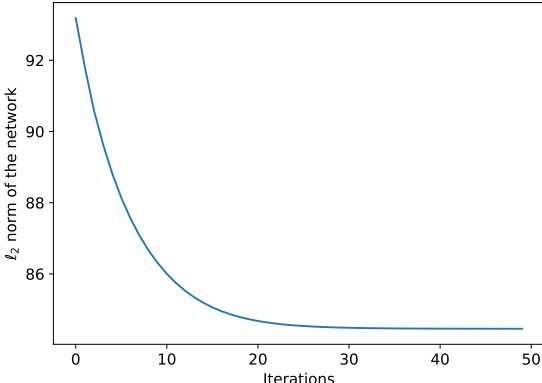

Figure 9: Iterating ENorm cycles on a randomly initialized ResNet-18 with no training.

We finally compare the evolution of the total $\ell_2$ norm of the network when training it, with or without ENorm. We use the setup described in Subsection 6.2 and use $p = 3$ intermediary layers. The results are presented in Figure 10. ENorm consistently leads to a lower energy level in the network.

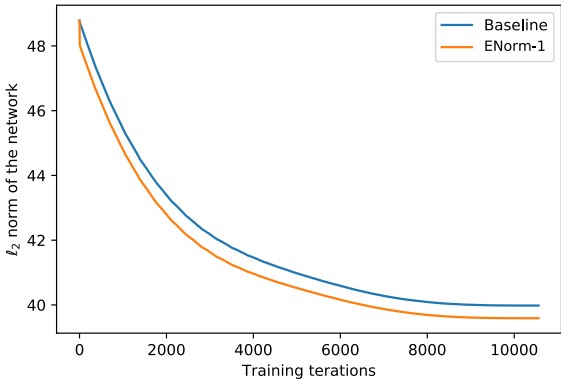

Figure 10: Training a fully-connected network on CIFAR-10, with (ENorm-1) or without (Baseline) Equi-normalization.

## E.2 ASYMETRIC SCALING: UNIFORM VS. ADAPTIVE

**MNIST auto-encoder.** For the uniform setup, we test for three different values of $c$, without BN: $c = 1$ (uniform setup), $c = 0.8$ (uniform setup), $c = 1.2$ (uniform setup). We also test the adaptive setup. The adaptive setup outperforms all other choices, which may be due to the strong bottleneck structure of the network. With BN, the dynamics are different and the results are much less sensitive to the values of $c$ (see Figures 11 and 12).

**CIFAR10 Fully Convolutional.** For the uniform setup, we test for three different values of $c$, without BN: $c = 1$ (uniform setup), $c = 0.8$ (uniform setup), $c = 1.2$ (uniform setup). We also test the adaptive setup (see Table 5). Once again, the dynamics with or without BN are quite different. With or without BN, $c = 1.2$ performs the best, which may be linked to the fact that the ReLUs are cutting energy during each forward pass. With BN, the results are less sensitive to the values of $c$.

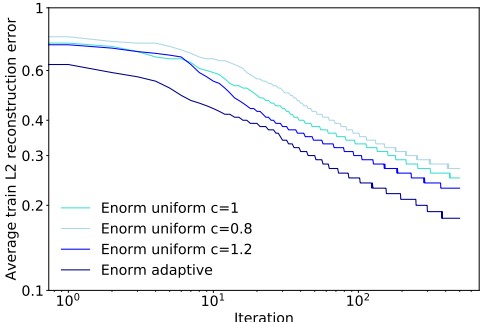

Figure 11: Uniform vs adaptive scaling on MNIST, without BN.

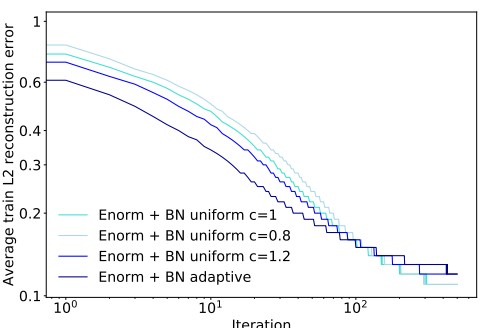

Figure 12: Uniform vs adaptive scaling on MNIST, with BN.

| Method | Test top 1 accuracy |
|---|---|
| ENorm uniform $c = 1$ | 86.98 |
| ENorm uniform $c = 0.8$ | 79.88 |
| ENorm uniform $c = 1.2$ | 89.31 |
| ENorm adaptive | 89.28 |
| ENorm + BN uniform $c = 1$ | 91.85 |
| ENorm + BN uniform $c = 0.8$ | 90.95 |
| ENorm + BN uniform $c = 1.2$ | 90.89 |
| ENorm + BN adaptive | 90.79 |

Table 5: Uniform vs adaptive scaling, CIFAR-10 fully convolutional

