# OpenReview forum: "Equi-normalization of Neural Networks"
_ICLR.cc/2019/Conference_

### Official Review · AnonReviewer3 · 2018-11-02

**Rating:** 5
**Confidence:** 3

**Review:**

The authors propose a new weight re-parameterization technique called Equi-normalization (ENorm) inspired by the Sinkhorn-Knopp algorithm. The authors show that the proposed method preserve functionally equivalent property in respect of the output of the functions (Linear, Conv, and Max-Pool) and show also that ENorm converges to the global optimum through the optimization. The experimental results show that ENorm performs better than baseline methods on CIFAR-10 and ImageNet datasets.

pros)
(+) The authors provide a theoretical ground.
(+) The theoretical analysis of the convergence of the proposed algorithm is well provided.
(+) The computational overhead reduced by the proposed method compared with BN and GN looks good.

cons)
(-) There is no comparison with other weight reparameterization methods such as Weight Normalization, Normalization propagation, Instance Normalization, or Layer Normalization.
(-) The evidence why functionally equivalence is connected to the performance or generalization ability is not clarified.
(-) The experimental results cannot consistently show the effectiveness of the proposed method in test accuracy. In Table 4, the proposed method outperforms BN, but In Table 2 and 3, BN is mostly better than the proposed method.
(-)  The batch size shown in Table 2 and 3 may be intended to show the batch-independent property of the proposed method, but BN is also doing well in those tables. Therefore, Table 2 and 3 are not adequate to show the batch-independent property.
(-) The proposed method should evaluate with deeper networks (e.g., ResNet-50, ResNet101, or DenseNet-169) to support the superiority over BN and GN.
(-) Adjusting c does not seem to be promising. In Table 2 and 3, ENorm-1 is better than ENorm-1.2, and also in Table 4, only the result of ENorm-1 is provided. The authors should do a parameter study with c to make all the experiments more convincing.

Comments)
- The experimental settings are not consistent. The authors should provide the reason why they set those settings or should include some studies about the parameters (for example about the paramter c).
- Section 3.7 is not clear to me.  How's the performance going on when adjusting c < 1?
- It is better for the authors to provide the Sinkhorn-Knopp algorithm (SK algorithm), which gave them an inspiration for this work, for better readability.
- Why eq.(4) is necessary? For iterative optimization? If so, the authors should incorporate a detailed explanation about this in the corresponding section.
- The authors should provide a detailed description of the parameter c. It is not clear why c is necessary, and please make sure the overall derivation does not need to be modified due to the emergence of c.
- It seems that the authors could compact the paper by highlighting key ideas.
- Typo: Annex A (on p.5).

The paper is written well and provides a sound theoretical analysis to show the main idea, but unfortunately, the experimental results do not seem to support the effectiveness of the proposed method.

---

> ### Author Response · Authors · 2018-11-14
> **Additional results that better demonstrate the performance of ENorm**
>
> We thank you for your clear and actionable feedback. Below are our detailed answers as well as additional results that prove the effectiveness of ENorm.
>
> - Parameter c (see Equation (11), that we now call "penalty coefficient"). We next provide a theoretically grounded derivation of our parameter. This derivation will replace subsection 3.7. Let us assume that we seek to minimize the global norm of the network weights, weighted by the number of elements in each matrix (W_k has size n_{k-1} x n_k):
>
> \varphi(\delta) = \sum_{k=1}^q \| \frac{1}{n_{k-1}n_k} D_{k-1}^{-1}W_kD_k \|_p^p
>
> Then, the same calculations as in Subsection 3.3 lead to new rescaling coefficients \tilde d_\ell{(r+1)} that are related to the standard rescaling coefficients d_\ell{(r+1)} by the following relation:
>
> \tilde d_\ell{(r+1)} = \sqrt{\frac{n_{k+1}}{n_{k-1}}} d_\ell{(r+1)}
>
> This gives a layer-dependent penalty coefficient c_k = \sqrt{\frac{n_{k-1}}{n_{k+1}}} that accounts for the architecture of the network. If the width of two consecutive layers is decreasing, c_k > 1 and if the width is increasing, c_k < 1. We empirically show that these coefficients consistently give the best results on the auto-encoder architecture on MNIST (see answer to Reviewer 2) and on the CIFAR10 fully connected and fully convolutional cases (see answer to Reviewer 1). Note that the convergence/uniqueness proof holds, because it amounts to replacing the weights W_k by their scaled counterparts \frac{1}{n_{k-1}n_k}W_k, thus preserving the assumptions made on the weights.
>
> - Comparison with WN and mix of ENorm + BN: see answer to Reviewer 1. We prove that ENorm + BN outperforms all other baselines, including Weight Norm. Note that without BN, the derivation of the penalty coefficient c_k presented above gives the best results compared to other scaling coefficients (we performed a “parameter study” as you rightly suggested). With BatchNorm, the optimal c is different and is cross validated (see also answers to Reviewers 1 and 2).
>
> - Functional equivalence connected to generalization: we rely on the following reasoning.
>          o Two networks are functionally equivalent if they provide the exact same output given the same input
>          o Two functionally equivalent networks provide exactly the same classification accuracy
>          o Two networks with rescaling equivalent weights are functionally equivalent, however their gradients wrt weights
>             are different, hence one step of SGD on the training set leads to different networks (not functionally equivalent) at
>             the next step even if they were functionally equivalent at the previous step
>          o The weights learned after several iterations of SGD depend on the criterion minimized over rescaling equivalent
>              weights at each iteration; depending on this criterion, we get different learned networks that have no reason to be
>             functionally equivalent. We can thus expect differences both in training accuracy but also in test / generalization
>             accuracy.
>          o It is however hard to predict theoretically if one criterion is better than another, this is more the object of empirical
>              evaluation, see answer to Reviewer 1.
>
> - The batch size dependency can be observed in our ImageNet results, as already shown in the paper “Group Normalization”. It is however a function of the number of classes in the dataset, thus its effect does not show up on CIFAR10. We will update the paper to clarify this point.
>
> - Other points. Equation 4 makes the proof in Appendix B clearer but is not essential for the main paper. We will move it in the Appendix to make the paper more concise. Also, we will add Sinkhorn’s algorithm in Appendix in an updated version of the paper

---

### Official Review · AnonReviewer2 · 2018-11-02
**A bit too incremental and needs comparison with Weight Normalization**

**Rating:** 7
**Confidence:** 4

**Review:**

Summary:
This paper introduces equi-normalization (ENorm): a normalization technique that relies on the scaling invariance properties of the ReLU, similarly to Path-SGD. Their method explicitly use this property to balance the weights of the network, without changing the function computed by the network. The main difference with Path-SGD is that the network is explicitly balanced, while Path-SGD uses a regularizer to implicitly balance the network. Since it doesn’t rely on mini-batch to normalize the network, Equi-normalization could be a good alternative to BN in small mini-batch regime. The method is validated on 3 tasks (MLP on CIFAR10, CNN on CIFAR10, Reidual Network on ImageNet).

Clarity:
The paper is quite clear, although a bit long (10 pages). The related work section is particularly nice. I really appreciated the “positioning” paragraph, which really explains how the method differs from others.

Novelty:
The paper is quite incremental, due to its similarities with Path-SGD. It seems quite close to what Weight Normalization is doing as well (see detailed comments).

Pros and Cons:
+ Clearly written
+ Clearly motivated
+ Nice review of literature
- Quite incremental (close to Path-SGD / Weight Normalization), and missing actual comparison with Weight Normalization, which seems to be the direct competitor of ENorm (see detailed comments)
- Some flaws in the experimental setup (see detailed comments), particularly in the fully-connected experiment.
- Doesn’t scale (yet) to deeper architectures, which is precisely where small batch sizes become a problem for BN and thus where ENorm would be needed.

Detailed Comments:
1. Differences with Weight Normalization:
I have trouble seeing the difference between the proposed method and Weight Normalization (WN), or the more advanced Normalization Propagation (NP). It seems that both methods are performing quite similar normalization: WN reparameterize the network such that ||W[:,j]||_2 = 1, so it seems that you wouldn’t need to re-balance the network when using WN. Could the authors elaborate on this? Moreover WN is simple to implement, fast, and also works in the small batches settings as well. Finally, since those methods are quite similar, the authors should compare their method against WN in their experimental setup.
2. Initialization of the Weights:
The Xavier initialization you are using in the CIFAR10 experiments is designed to work with Tanh activations functions. For ReLUs, one should use the Kaiming initialization, which has been proven to work way better for ReLUs (He et al., 2015a).  This certainly explains the poor performances of your baseline when you increase the number of layers in Figure 4, and probably explains why you need to add a BN layer at the end of your network to help with the training of the baselines. I suggest the authors to re-run the baselines using proper initialization for ReLUs.
3. Fully-Connected Layers Benchmark:
I think the fully-connected benchmark you used is quite poor. A baseline with 1 layer only reaches ~54 % test accuracy and your method needs a 11 layer model to increase this baseline performance of about ~0.5 % only. The deep autoencoder on MNIST (see e.g. Desjardins et al., Natural neural networks, NIPS 2015), would probably be a better benchmark for fully-connected layers (of course using ReLUs in place of Sigmoids).  It would also reinforce your empirical results by adding a 3rd dataset.
4. ImageNet Experiment:
Do you use Ghost Batch Normalization in this experiments (i.e. calculating the BN statistics on each GPU separately)? It would certainly explain the poor performances of BN with tiny batch size (32 examples on 8 GPUs means only 8 examples per GPU).
5. Computation Time performances:
It is stated in the conclusion that “using ENorm during training adds a much smaller computational overhead than BN or GN …”. I can see that Table 1 gives an overview of the number of elements accessed during normalization, but I do think that a proper plot showing the accuracy versus the wall-clock time would be a better way of showing how your method compares in practice with BN or GN. Moreover, as stated previously, comparison with WN (and / or NP) need to be performed as well, especially because WN and NP are also way faster to compute than BN and GN.
6. Shortening the paper:
The recommended paper size for ICLR is 8 pages. Here would be a few pointers that could help you reduce a bit the length of the paper: Introduction and Related Work takes up 3 pages already and I think there is quite some overlap between the 2 (about BN in particular), so there is probably quite a lot of space to gain here if the authors were to reduce a bit the introduction or make the related work a sub-section of the introduction. The ENorm presentation is 4 pages long (which is quite a lot). Section 3.6 might be totally discarded since it is vanilla application of the chain rule. The extension to convolution layers and max pooling could be transferred to the appendix.

Minor Comments:
You should add “Optimizing neural networks with kronecker-factored approximate curvature” in the literature review about optimization landscape, as it is an important research direction on natural gradient.

Conclusion:
All in all, I find that this work is a bit too incremental, missing some important comparisons with other techniques and its experimental setup could definitely be improved. Also, the speedup claims should also be supported with empirical experimentation.

Revision:
I thank the authors for the all the extra experiments they performed. The paper looks good to me, and increased my evaluation accordingly.

---

> ### Author Response · Authors · 2018-11-14
> **Additional results that better demonstrate the performance of ENorm**
>
> We thank you for your thorough and very detailed review. We provide below our answers to all the questions you raised as well as updated results that demonstrate the effectiveness of ENorm thanks to your suggestions.
>
> - Positioning. We thank you for appreciating our Positioning paragraph and hope to make it even clearer with the following remarks. We state the main differences between ENorm, Path-SGD and WeightNorm (WN). First, while ENorm and Path-SGD both seek to balance the network, one normalization is explicit and the other implicit. Second, it is unclear how to extend Path-SGD to convolutional or even ResNet-like architectures (the authors restrict themselves to the fully-connected case in their papers), while ENorm can be readily adapted to the convolutional case and even to ResNet-like architectures. Besides, WN parameterizes the weights in a polar-like manner, w = g * v /|v|, where g is a learnt scalar and v is the learnt weights. While requiring mean-only BN and greedy layer-wise initialization, WN does not “balance the network” but rather individually scales each layer.
>
> - Xavier Initialization: You are right to mention that Kaiming He’s initialization is now widely adopted. As you suggested, we re-ran all our baselines with this initialization and provide our results in the next paragraph as well as in the answers to Reviewer 1. Note that BN allows one to be less careful about the initialization, as documented in the original BN paper. Thus, changing the initialization did not affect the performance of setups involving BN.
>
> - Ghost BN:  as mentioned in our paper, we follow exactly Kaiming He’s setup in Group Normalization. Indeed, in their paper, they state: "As standard practice, we use 8 GPUs to train all models, and the batch mean and variance of BN are computed within each GPU". Besides being common practice, this setup avoids to significantly slow down the training, as synchronizing the mean and variance across all 8 GPUs during each forward is time consuming.
>
> - Theoretical time gain: as mentioned in our paper, empirically observing the speedup of ENorm would require writing our own cuda kernels instead of relying on the high-level PyTorch library. We conducted an additional experiment to demonstrate the speedup gain achievable with such kernels. We report the training speed (forward + backward time) of a ResNet18 on Imagenet with a Quadro GP100 and a batch size of 256:
>          o 2,000 samples/s with the built-in PyTorch implementation of Batch Norm (relying on custom cuda kernels)
>          o 850 samples/s with our own implementation of Batch Norm using high level PyTorch functions
>
> Here are our additional results.
> - CIFAR10 fully-connected and fully convolutional: see answer to Reviewer 1. We demonstrate that ENorm + BN outperforms all other baselines, including Weight Norm combined with BN.
>
>  - MNIST AutoEncoder. We thank you for suggesting another evaluation setup. We demonstrate that ENorm + BN outperforms all other methods, including Weight Norm.
>          o Setup. As in the paper you mentioned, the encoder has the structure FC(784, 1000)-ReLU-FC(1000, 500)-ReLU-
>             FC(500, 250)-ReLU-FC(250, 30) and the decoder has the symmetric structure. We use Kaiming’s initialization for
>            the  weights. We select the learning rate in {0.001, 0.01, 0.1} based on the training error and decay it linearly to
>            zero.
>          o Scaling coefficient. Note that following Reviewer 3’s suggestions, we derived a theoretically grounded formula
>             for our asymmetric penalty coefficient c, that now depends on the layer sizes. Here, the optimal scaling vector
>             following this formula (see answer to Reviewer 3) is c* = (1.25, 2, 4.08, 1, 0.24, 0.5, 0.80). We compare it against
>            c1 = (1, 1, 1, 1, 1, 1, 1), c2 = (1.2, 1.2, 1.2, 1, 0.8, 0.8, 0.8, 0.8) and c3 = (0.8, 0.8, 0.8, 1, 1.2, 1.2, 1.2).
>          o Results. We present the results (average training reconstruction error after 500 epochs) below. Those results are
>             consistent across whole training iterations and we will update the paper with plots of the training loss. ENorm +
>             BN  (with c = c2 here) outperforms all other baselines, while the choice of c* for ENorm alone significantly
>             improves the performance of the method.
>
> Method                       Average L2 training loss
> ----------------------------------------------------------------
> Baseline                        0.808
> Path-SGD                      0.801
> BN                                  0.171
> WN + BN                       0.162
> ----------------------------------------------------------------
> ENorm c*                     0.179
> ENorm c1                     0.251
> ENorm c2                     0.232
> ENorm c3                     0.278
> -----------------------------------------------------------------
> ENorm + BN                0.102

---

### Official Review · AnonReviewer1 · 2018-11-02
**Good method and theoretical contribution, but not convincing results**

**Rating:** 7
**Confidence:** 3

**Review:**

The authors propose a new regularization method for neural networks. The main idea is to reparametrize the neural network after each update by rescaling the weights, without changing the encoded function. The proposed algorithm is proved to converge to a unique canonical representation of the weights.

The paper is clear and well written. The proof structure in the appendix seems coherent even though I haven't checked all the details.  Moreover, the authors detail the application of the method to all the different building blocks of modern architectures.

Up to my knowledge, the idea is novel. Moreover, it can have a high impact on the robustness of training. However, the results are somewhat disappointing.  While the authors present the method as an alternative to batch normalization, most of the reported results show a better performance for BN.

One of the drawbacks of batch normalization is it's incompatibility with other regularizers such as Dropout. Did the authors try to combine ENorm with Dropout?
Another direction that can be worth to investigate, in the same space of improving the robustness of training, is to try to combine this reparametrization with natural gradient updates.
Another question that remains open is the following: Even though the algorithm converges to a unique minimizer, it is not guaranteed that the obtained minimizer is good. Indeed, the authors note in their discussion that the criterion they optimize might be not optimal.

To summarize, the idea and theoretical contribution is significant, but the work can be improved.

==================
After rebuttal
==================
The authors provided new experiments supporting the proposed method. I am happy to increase my rating.

---

> ### Author Response · Authors · 2018-11-14
> **Additional results that better demonstrate the performance of ENorm**
>
> We thank you for your feedback and for recognizing both the theoretical contribution and the potential impact this work can have in the robustness of training. We next answer your specific questions and provide additional results that better demonstrate the performance of ENorm.
>
> In the paper, we focused on evaluating our method independently from other normalization methods (e.g. BatchNorm or Dropout). However, following your suggestion, we additionally:
>              (1) studied the combination of ENorm and other normalization methods
>              (2) provide a comparison with Weight Norm (also provided after a comment posted on Oct 1 on OpenReview). Note that Weight Norm does require mean-only BN to get competitive results with BN, as well as a greedy layer-wise initialization as mentioned in the original paper (Figure 2). This is in contrast to all the other methods presented in the paper.
>
> Regarding the criteria we optimize for, you are right to mention that it is yet not known (up to our knowledge) if it is optimal, and that we specifically made this comment in our concluding remarks. Rather than “optimality” (which is an absolute goal and hard to assess) we next provide some empirical insights of why the criteria we optimize for (the global norm of the weights) is “reasonable”. Indeed, as mentioned in the Path-SGD paper (2015, Figure 1), optimizing an (artificially) unbalanced network is hard. As ENorm explicitly balances the network (see Appendix A, Figures 6 and 7), it circumvents such optimization problems and leads to a good optimization trajectory.
>
> Additional results (that we will incorporate in an updated version of the paper)
> - CIFAR10 fully connected. Following Reviewer 3’s suggestions, we derived a theoretically grounded formula for our asymmetric penalty coefficient c, that now depends on the layer sizes. Here,  the optimal penalty vector (see answer to Reviewer 3) is c* = (2.48, 1, …, 1, 7.07). We compare it against c1 = (1, …, 1), c2 = (1.2, …, 1.2) and c3 = (0.8, …, 0.8). We use c = (1.2, …, 1.2) for ENorm + BN. Therefore, when stacking 11 layers or more, our method outperforms all others by a significant margin when combined with BN.
>
> Method \ Layers           3             7           11             15           19
> -----------------------------------------------------------------------------------
> Baseline                 53.03      52.91      51.66      51.40      50.49
> BN                           52.24      52.57      51.73      51.78      51.69
> GN                           53.32     53.56       53.85      53.11      52.58
> WN + BN                 54.86     54.14       53.91      53.49      52.50
> -----------------------------------------------------------------------------------
> ENorm c*               52.09      53.16      53.76      52.49      52.17
> ENorm c1               53.20      53.43      52.19      50.70      51.82
> ENorm c2               53.31      53.74      54.04      53.52      52.70
> ENorm c3               53.15      50.18      43.02      38.26      34.89
> ------------------------------------------------------------------------------------
> ENorm + BN          53.05       53.61     54.53       54.95      55.04
>
> - CIFAR10 fully convolutional. We use a batch size of 128 and a momentum of 0.9, and cross-validate the other hyper-parameters as stated in the paper. Here, the optimal penalty vector following our formula (see answer to Reviewer 3) is c* = (0.15, 1, 1.41, 0.7, 1, 1.41, 0.7, 1). We compare it against c1 = (1, 1, 1, 1, 1, 1, 1, 1), c2 = (1.2, 1.2, 1.2, 1.2, 1.2, 1.2, 1.2, 1.2) and c3 = (0.8, 0.8, 0.8, 0.8, 0.8, 0.8, 0.8, 0.8). For ENorm + BN, we use c = (1.2, 1.2, 1.2, 1.2, 1.2, 1.2, 1.2, 1.2). Again, the combination of ENorm + BN gives the highest accuracy.
>
> Method                       Test top1 accuracy
> -----------------------------------------------------------------
> Baseline                       88.94
> BN                                 90.58
> GN                                 90.36
> WN + BN                       90.50
> -----------------------------------------------------------------
> ENorm c*                     89.28
> ENorm c_1                    86.98
> ENorm c_2                    89.31
> ENorm c_3                    79.88
> ------------------------------------------------------------------
> ENorm + BN                 91.85
>
> - ENorm + Dropout: early experiments on the fully-connected setup on CIFAR10 show no improvement when adding Dropout (with a drop probability p=0.5).
> - Natural gradient updates: we leave this interesting direction for future work.

---

### Public Comment · (anonymous) · 2018-10-01
**Questions of experiments and related work**

Normalizing CNNs without depending on batch statistics is very important and interesting.  I have a few questions. Why not compare with WN in experiments? In the WN paper, WN achieves comparable performance to BN when additional regularization is properly introduced such as mean-only BN or dropout. Does additional regularization help ENorm?
A related work switchable normalization is missing.  It would be interesting to see how does it compare to BN, GN, and ENorm in small-scale problem like CIFAR-10.

---

> ### Author Response · Authors · 2018-10-02
> **Additional experiments and answer**
>
> Thank you for your constructive feedback. Here is our detailed answer:
>
> (1) Weight Normalization. It is an interesting method that we cite in our work. We have performed additional experiments and report below the results of WN + mean-only BN (as described in the original paper, using the WN implementation distributed with PyTorch). We use the exact same CIFAR-10 fully-connected setup as described in the article, except that we cross-validate the learning rate among the larger set {0.0005, 0.001, 0.005, 0.01, 0.05, 0.1, 0.5}. Note that we additionally perform a layer-wise initialization for WN, as described in the original paper, to obtain competitive results. We did not perform any layer-wise initialization for all the other methods presented in the paper. We give the ENorm-1.2 and BN results for reference. Note that ENorm is more successful than WN+mean-only BN for training deep fully connected network.
>
> Intermediate layers  	| 1		3		5		7		9		11		13		15		17		19 	    |
> ------------------------------	| ---------------------------------------------------------------------------------------------------------------------|
> Baseline                        	| 54.07	53.65	53.63	53.20	53.35	52.32	52.01	50.47	47.27	40.81   |
> ENorm 				| 53.92	54.35	54.56	54.59	54.67	54.70	54.48	53.80	53.42	52.88   |
> WN + mean-only BN	| 55.13	55.15	54.80	54.24	53.86	53.90	53.11	53.27	51.32	51.18   |
> Delta				| -1.21	 -0.80 	-0.24 	+0.35 	+0.81 	+0.80 	+1.37  	+0.53 	+2.10 	+1.70   |
>
> (2) Switchable normalization. The idea to learn attention weights to select among different ways to compute the activation statistics (BN, GN, IN, etc.) is interesting. We will cite the reference in our Related Work section in an updated version of our paper, and argue that while we focus on evaluating normalization methods as independently as possible, an aggregation/combination of normalization methods into one "meta-normalization" scheme is a topic for further research. Early additional experiments suggest that applying ENorm with mean-only BN (for example), without balancing the BN weights, leads to improvements over ENorm in the CIFAR-10 fully-connected case.

---

### Author Response · Authors · 2018-11-26
**Updated paper with additional experiments showing the effectiveness of ENorm**

Dear reviewers,

We would like to thank you again for your constructive feedback. Following your suggestions, we updated our paper with the main following modifications:

- Additional results for CIFAR10 fully-connected and fully convolutional with comparison to WN
- New experimental setup: auto-encoder on MNIST with comparison to WN
- Results for ENorm + BN that outperforms all other methods in 3 setups, including WN + BN
- Proper definition of the asymmetric scaling (Subsection 3.6, two "natural" setups for defining layer-wise parameters c_k)
- A parameter study for different values and setups of asymmetric scaling
- We have shortened the paper with a more concise Section 3 and some parts of Section 4 moved in Appendix
- A clearer positioning with respect to WN
- We added suggested references in the Related Work section

Note that all the modifications on the paper are directly linked to a comment of at least one reviewer. To facilitate the reading, the main modifications compared to the original paper are highlighted in blue.

As stated in the paper, we will make the code to reproduce our experiments publicly available. We hope those results better demonstrate the effectiveness of ENorm as well as the theoretical contribution of the paper.

---

### Public Comment · (anonymous) · 2018-12-06
**Questions for Reproducibility Challenge**

Hello, we are working on replicating some the results from this paper as part of the ICLR 2019 Reproducibility Challenge. We are focusing on reproducing the results of the CIFAR-10 fully-connected network, and we had a few questions that we would like to ask:

- Do you use a biasing term during your experiments? If so, how is the biasing term initialized?
- What was the activation function used for the final layer?
- Are baseline results obtained with the exact same setup as ENorm, but without any normalization?
- Throughout the training process, did you use the validation accuracy to determine the best set of weights for evaluating the test accuracy, or simply trained for 60 epochs and used the final weights?
- What are the learning rates that provided you with the best validation accuracy?
- Could you provide us with numerical values (table) you used to plot your data? We are particularly interested in having the values for the baseline and ENorm cases.
- We noticed that these values were given in the comments section below, but that they seem to have changed with the new version of the paper. Could you also explain what caused that difference?
-We are having stability issues during training. Particularly, at higher intermediary layer counts, the loss is NaN after a single epoch. Do you have any insight on what could cause this and how to stabilize the process?
-Do you decay the learning rate over the training process?

Any feedback that you can provide us would be greatly appreciated!

---

> ### Author Response · Authors · 2018-12-07
> **Code for reproducing our experiments**
>
> Thanks for your interest in our work!
>
> - We provide the code to reproduce the experiments you are working on as a .zip folder at the anonymous url indicated below. The folder contains a script "reproduce.sh" that (1) precisely describes all the hyper parameters that were found to work best on the validation set and (2) provides the numerical values we obtained.
>
> - We provide below some general answers and refer you to the code for all the details:
>     o The stability issues (NaN losses) you encounter may come from the fact we do not put a non-linearity after the last
>         layer (right before the softmax), as is the case for all DNN architectures.
>     o We cross-validate on a separate validation set and report the accuracies on the test set obtained with the final
>         weights (at the end of the training, averaged over 5 independent runs with the optimal cross-validated parameters)
>         as it is a common practice.
>     o The values in the paper are either (1) coming from an extended range of hyper-parameters or (2) averages over 5
>         times. This explains the difference in the numbers. Additionally, in the initial submission, we used Xavier initialization
>         for the weights and have re-run all the experiments using Kaiming's initialization, as requested by the reviewers.
>     o ENorm and the baseline case are evaluated using exactly the same setup.
>
> We would be happy to assist in the reproducibility of our work and answer all the further questions you may have.
>
> Anonymous code url: https://drive.google.com/file/d/1uWMwvzIrcuHA2ZVl6IUa80_CDK9HFLdR/view?usp=sharing

---

### Author Response · Authors · 2019-02-22
**Final version**

We uploaded the final version of the paper.

We would like to thank the reviewers for their insightful comments which helped us to significantly improve the paper’s clarity and the experimental validation, as well as the AC for their detailed meta-review.

See you in New Orleans!

---

### Meta-Review · Area_Chair1 · 2018-12-09
**neat normalization method, well-executed**

**Confidence:** 5
**Recommendation:** Accept (Poster)

**Metareview:**

The proposed ENorm procedure is a normalization scheme for neural nets whereby the weights are rescaled in a way that minimizes the sum of L_p norms while maintaining functional equivalence. An algorithm is given which provably converges to the globally optimal solution. Experiments show it is complementary to, and perhaps slightly better than, other normalization schemes.

Normalization issues are important for DNN training, and normalization schemes like batch norm, weight norm, etc. have the unsatisfying property that they entangle multiple issues such as normalization, stochastic regularization, and effective learning rates. ENorm is a conceptually cleaner (if more algorithmically complicated) approach. It's a nice addition to the set of normalization schemes, and possibly complementary to the existing ones.

After a revision which included various new experiments, the reviewers are generally happy with the paper. While there's still some controversy over whether it's really better than things like batch norm, I think the paper would be worth publishing even if the results came out negative, since it is a very natural idea which took some algorithmic insight in order to actually execute.